

# Quantification of fossil fuel $CO_2$ from combined CO, $\delta^{13}CO_2$ and $\Delta^{14}CO_2$ observations

Jinsol Kim[1], John B. Miller[2], Charles E. Miller[3], Scott J. Lehman[4], Sylvia E. Michel[4], Vineet Yadav[3], Nick E. Rollins[1], and William M. Berelson[1]

[1]Department of Earth Sciences, University of Southern California, Los Angeles, CA 90089, USA
[2]Oceanic and Atmospheric Administration Global Monitoring Laboratory, Boulder, CO 80305, USA
[3]Jet Propulsion Laboratory, California Institute of Technology, Pasadena, CA 91109, USA
[4]Institute of Arctic and Alpine Research, University of Colorado, Boulder, CO 80309, USA

*Correspondence to*: Jinsol Kim (jinsolki@usc.edu)

**Abstract.** We present a new method for partitioning observed $CO_2$ enhancements ($CO_2$xs) into fossil and biospheric fractions (Cff and Cbio) based on measurements of CO and $\delta^{13}CO_2$, complemented by flask-based $\Delta^{14}CO_2$ measurements. This method additionally partitions the fossil fraction into natural gas and petroleum fractions (when coal combustion is insignificant). Although here we apply the method only to discrete flask air measurements, the advantage of this

method (CO and $\delta^{13}CO_2$-based method) is that $CO_2$xs partitioning can be applied at high frequency when continuous measurements of CO and $\delta^{13}CO_2$ are available. High frequency partitioning of $CO_2$xs into Cff and Cbio has already been demonstrated using continuous measurements of CO (CO-based method) and $\Delta^{14}CO_2$ measurements from flask air samples. Relative to calculating $CO_2$ff directly from $\Delta^{14}CO_2$, we find that the uncertainty in $CO_2$ff estimated from the CO and

$\delta^{13}CO_2$-based method averages 3.2 ppm which is significantly less than the CO-based method which has an average uncertainty of 4.8 ppm. Using measurements of CO, $\delta^{13}CO_2$ and $\Delta^{14}CO_2$ from flask air samples at three sites in the greater Los Angeles region, we find large contributions of biogenic sources that vary by season. On a monthly average, the biogenic signal accounts for −14 to +25 % of $CO_2$xs with larger and positive contributions in winter and smaller and negative

contributions in summer due to net respiration and net photosynthesis, respectively. Partitioning $CO_2$ff into petroleum and natural gas combustion fractions reveals that the largest contribution of natural gas combustion generally occurs in summer, which is likely related to increased electricity generation in LA power plants for air-conditioning.



## 1 Introduction

The world's cities account for up to 70 % of global greenhouse gas (GHG) emissions, while covering less than 2 % of the Earth's surface (IPCC, 2014). Cities around the world have started implementing mitigation strategies to reduce carbon dioxide ($CO_2$) emissions and collaborate with

35 each other in organizations such as the C40 Cities Climate Leadership Group (https://www.c40.org/) and the Global Covenant of Mayors for Climate and Energy (https://www.globalcovenantofmayors.org/). To support urban efforts, monitoring systems are necessary to evaluate and verify reductions attributable to specific mitigation strategies.

Current understanding of anthropogenic $CO_2$ emissions mainly derives from methods that estimate

aggregate emissions in a domain using economic statistics such as total fuel sales. These "bottom-up" methods provide specific location and process information that rely on mapping the source-specific emission factors and measurements of activities (e.g., McDonald et al. 2014; Gurney et al. 2019; Gately and Hutyra 2017). In contrast, more recently "top-down" methods that quantify emissions from measurements of atmospheric $CO_2$ have been used to estimate emissions. These

top-down approaches typically use either a mass balance technique where an initial estimate is not required (e.g., Heimburger et al. 2017; Ahn et al. 2020) or an inverse/data assimilation approach where observations and a prior map of emissions are combined to generate a best estimate (e.g., Sargent et al. 2018; Turner et al. 2020; Lauvaux et al. 2016, 2020).

To estimate anthropogenic $CO_2$ emissions using top-down method, it is crucial to separate the

50 fossil fuel signals from the biogenic signals, which can vary from negative (uptake) to positive (emission) across the annual cycle. Recent analyses of urban $CO_2$ suggest that biogenic emissions and uptake have significant magnitudes relative to fossil fuel fluxes, especially during the growing season (Sargent et al. 2018; Vogel et al. 2019; Miller et al. 2020). Previous top-down studies used biosphere models to estimate biogenic fluxes and then determine the balance of emissions

attributable to fossil fuel combustion (Sargent et al. 2018; Turner et al. 2020; Lauvaux et al. 2020). However, even with recent improvements in biosphere models (Wu et al. 2021; Gourdji et al. 2022) the actual magnitude and variability of these fluxes are still not well constrained (Hardiman et al. 2017), potentially leading to unknown observational bias in the associated estimates of fossil fuel derived emissions.

Radiocarbon ($^{14}CO_2$) provides the ability to separate biogenic and anthropogenic $CO_2$ fluxes and mole fractions from an observational point of view (e.g. Levin, 2003; Turnbull 2006).



Observational methods rely on the fact that fossil fuels and the resultant $CO_2$ produced during combustion are completely devoid of $^{14}C$ (i.e., $\Delta^{14}C_{ff}$ = -1000‰ on the widely used Delta scale). Measurements of $\Delta^{14}CO_2$, acquired at time scales of weeks to months, allow quantification of

seasonal variations in biogenic and fossil contributions to the atmospheric $CO_2$ mole fraction (e.g., Djuricin et al. 2010; Miller et al. 2012; Turnbull et al. 2015). $^{14}C$ methods typically require air sample collection, preparation and analysis via accelerator mass spectrometry which limits the number of measurements, although a number of promising optical methods for in situ $^{14}CO_2$ measurement at natural abundance are currently being developed (Fleisher et al. 2017; Genoud et

al. 2019; McCartt and Jiang 2022).

On the other hand, carbon monoxide (CO) is a widely used tracer that can be measured continuously in situ using high-precision optical analyzers (e.g., Vogel et al. 2010; Newman et al. 2013; Turnbull et al. 2015). CO is often co-emitted with fossil fuel $CO_2$ ($CO_2ff$) during incomplete combustion. If the $COxs:CO_2ff$ ratio ($R_{ff}$, where COxs is the CO enhancement above background)

is well constrained, continuous CO measurements combined with $R_{ff}$ can provide an estimate of continuous $CO_2ff$. However, this approach is challenging because $R_{ff}$ at a site may vary significantly on timescales ranging from hours to years (Levin and Karstens 2007; Vogel et al. 2010). $CO:CO_2$ emission ratio can vary by sources depending on the carbon content of the fuel and combustion conditions. Due to the impacts of atmospheric transport at a given observation site

and the variability in the source combination in time and space, $R_{ff}$ also varies in time and space. Vardag et al. (2015) proposed dividing fossil fuel emissions further into two groups that may display less variability in $CO:CO_2$ emission ratio. If one group is well constrained by CO and the other by $^{13}CO_2$, each group can be identified by combining CO and $^{13}CO_2$ observations. Vardag et al. focused on separating traffic from non-traffic emissions, or biofuel emissions from the other

fossil fuel emissions. However, no significant benefit of combining CO and $^{13}CO_2$ was found because traffic and biofuel $CO_2$ do not produce distinct $CO:CO_2$ emission ratio or $^{13}CO_2$ isotopic signatures compared to the other $CO_2ff$ source terms.

Here, we differentiate $CO_2$ signals from biogenic, petroleum and natural gas sources by combining CO, $\delta^{13}CO_2$, and $\Delta^{14}CO_2$ measurements. The combination of $\Delta^{14}CO_2$ and $\delta^{13}CO_2$ has been used

previously to distinguish biogenic, petroleum and natural gas signals for air sampling events (Lopez et al. 2013; Djuricin et al. 2010) and at seasonal scale (Newman et al. 2016). In contrast, the combination of CO and $\delta^{13}CO_2$, which can both be measured at high frequency, enables source



partitioning at higher temporal resolution. We demonstrate the agreement between the existing $\Delta^{14}CO_2$, $\delta^{13}CO_2$ and newly proposed CO, $\delta^{13}CO_2$ methods. This establishes the utility of the CO

and $\delta^{13}CO_2$ in partitioning $CO_2$xs into fossil fuel and biogenic components, with further partitioning of fossil fuel sources into petroleum and natural gas sources, in the Los Angeles megacity.

## 2 Methods

Here, we describe two methods for separating fossil fuel and biogenic components from

100 atmospheric $CO_2$ measurements in the complex urban environment of the Los Angeles megacity (LA). Section 2.2 describes our application of the method already described by Newman et al. (2016) using $\Delta^{14}CO_2$ and $\delta^{13}CO_2$ observations. The details of the new method utilizing CO and $\delta^{13}CO_2$ measurements are described in section 2.3. Briefly, we take advantage of the fact that the combination of the $CO:CO_2$ emission ratio and the $^{13}CO_2$ isotopic signature reveal a very distinct

pattern for biogenic, petroleum and natural gas sources. However, this approach requires knowledge of the $CO:CO_2$ emission ratio and the isotopic signature of each source. We apply isotopic signatures reported by previous studies, and $CO:CO_2$ emission ratios are determined for LA using measurements of CO, $\delta^{13}CO_2$ and $\Delta^{14}CO_2$ from flask samples. Flask measurements are described in section 2.1 and the source apportionment from $\Delta^{14}CO_2$ and $\delta^{13}CO_2$ observations,

which is used to derive $CO:CO_2$ emission ratios for each source, is described in section 2.2.

### 2.1 Measurements

We use measurements from air samples collected at 2 p.m. local standard time at three existing Los Angeles Megacity Carbon Project sites: University of Southern California (USC), California State University, Fullerton (FUL), and Granada Hills (GRA) (Miller et al. 2020). Air samples were

115 collected from November 2014 to March 2016 using National Oceanic and Atmospheric Administration (NOAA) programmable flask packages (PFPs) and programmable compressor packages (Sweeney et al. 2015). The samples were sent back to the NOAA Global Monitoring Laboratory where greenhouse gases including $CO_2$ as well as CO mole fractions were measured using NOAA's high-precision/high-accuracy greenhouse gas measurement system (Sweeney et al.

2015). After the measurement, residual air is extracted from PFP flasks and $CO_2$ is isolated for $^{14}C$





measurement using established cryogenic and mass spectrometric techniques (Lehman et al. 2013). Samples are purified, graphitized and packed into individual targets at the University of Colorado, Boulder, Institute of Arctic and Alpine Research (INSTAAR) and then sent to the University of California, Irvine, Keck Accelerator Mass Spectrometry Facility for high- precision $\Delta^{14}C$

measurement. One-sigma measurement uncertainty is ~1.8 ‰, equivalent to ~1.2 parts per million (ppm) of recently added fossil fuel–$CO_2$. $\delta^{13}CO_2$ in PFP samples is measured by dual inlet isotope ratio mass spectrometry with a precision of approximately 0.02 ‰ at the INSTAAR Stable Isotope Laboratory (Vaughn et al. 2004; Sweeney et al. 2015)

Enhancement of each species is defined relative to a time-dependent background level, which is

based on nighttime (2 AM local standard time) measurements made at Mount Wilson Observatory (MWO; Fig. 1) located at 1,670 m above sea level. Nighttime air at MWO generally represents the relatively clean, well-mixed free troposphere since polluted LA Basin boundary layer air has typically descended back into the basin by this time. After an additional step of filtering obvious outliers corresponding to pollution events indicated by anomalously elevated values were

interpolated to the time of observations within the LA Basin by fitting curves to the screened MWO data (Fig. 2). A further analysis of associated CO measurements indicates that background reconstructed using nighttime air samples from MWO is representative of clean background air coming from either on- or off-shore (Miller et al. 2020).

## 2.2 Partitioning $CO_2$ signals using flask-based $\Delta^{14}CO_2$ and $\delta^{13}CO_2$ measurements

Our general approach to distinguishing $CO_2$ signals from biogenic, petroleum and natural gas sources using $\Delta^{14}CO_2$ and $\delta^{13}CO_2$ follows the procedure described by Newman et al. (2016). Following previous derivations (e.g., Turnbull et al, 2006; Miller et al. 2020), we start with the definition for $CO_2ff$ which is based on mass balances for the atmospherically conserved quantities $\Delta^{14}C \times CO_2$ and $CO_2$:

$$C_{ff} = \frac{C_{obs}(\Delta_{obs} - \Delta_{bkg})}{(\Delta_{ff} - \Delta_{bkg})} - \frac{C_r(\Delta_r - \Delta_{bkg})}{(\Delta_{ff} - \Delta_{bkg})} \qquad (1)$$

Measured $CO_2$ mole fractions and $\Delta^{14}C$ values are abbreviated as $C$ and $\Delta$. Subscripts 'obs', 'bkg', 'ff' and 'r' represent observations, background, fossil fuel, and respiration, respectively. $\Delta_{ff}$ is equal to $-1000$ ‰. As in the Miller et al. (2020) study focusing on LA, we estimate the value of





the small respiratory term, $-C_r(\Delta r - \Delta bkg)/(\Delta ff - \Delta bkg)$, as 0.25 ppm. The overall uncertainty of

$C_{ff}$ for LA measurements during 2015 is approximately 1.2 ppm, which includes 100% uncertainty assigned to the respiratory term. $C_{ff}$ and $C_{bio}$ ($C_{bio} = C_{xs} - C_{ff}$) are calculated for all available flask air samples during the 2014 – 2016 sampling period, a frequency of approximately three times per week at each of the three sites.

$C_{ff}$ is then separated into signals from petroleum and natural gas combustion using $^{13}C:^{12}C$ ratios

($\delta^{13}C$ as defined by standard isotopic definition) measured on the same air samples. First, the flux weighted-mean $\delta^{13}C$ signature of all sources located in the observation footprints ($\delta_{src}$) is determined on a sample-by-sample basis using the combined mass balances for $\delta^{13}C \times CO_2$ and $CO_2$:

$$\delta_{src} = \frac{\delta_{obs} \times C_{obs} - \delta_{bkg} \times C_{bkg}}{C_{obs} - C_{bkg}} \qquad (2)$$

where $\delta$ is short-hand for $\delta^{13}CO_2$. The uncertainties in $C_{obs}$, $C_{bkg}$, $\delta_{obs}$, and $\delta_{bkg}$ are 0.1 ppm, 1.5 ppm, 0.02 ‰, and 0.08 ‰, respectively. The "obs" uncertainties are measurement uncertainties, while the "bkg" uncertainties are determined as the standard deviation of the difference between the observations and their smoothed curve representation at MWO. The median uncertainty in $\delta_{src}$ is 3.0 ‰ and is calculated by propagating the uncertainties listed above, including covariance

between $\delta^{13}C$ and $\delta^{13}C \times CO_2$.

We combine $C_{ff}$ (eq. 1) and $\delta_{src}$ (eq. 2) to determine the $\delta^{13}C$ signature of fossil fuel emissions, $\delta_{ff}$:

$$\delta_{src} = \delta_{ff} \times f_{ff} + \delta_{bio} \times (1 - f_{ff}) \qquad (3)$$

Rearranging yields:

$$\delta_{ff} = \frac{\delta_{src} - \delta_{bio} \times (1 - f_{ff})}{f_{ff}} \qquad (4)$$

where $f$ is the fraction. Following Newman et al. (2016), we take the isotopic signature of biospheric $CO_2$ fluxes ($\delta_{bio}$) to be $-26.6 \pm 0.5$ ‰ based on the analysis of Northern Hemisphere mid-latitude $CO_2$ and $\delta^{13}C$ observations (Bakwin et al. 1998), which reflects the predominance of $C_3$ photosynthesis. However, because LA turfgrasses, which could account for a significant

fraction of urban $CO_2$ fluxes [Miller, 2020], are often $C_4$ species (e.g., Bermuda and Buffalo



grasses), we also conduct tests using $\delta_{bio} = -20$ ‰, representing a C3/C4 mix (see Fig. S1). $f_{ff}$ is the fraction of $C_{ff}$ in $C_{xs}$, i.e., $C_{ff}/C_{xs}$, and $1 - f_{ff} = f_{bio}$. Lastly, the proportion of $C_{ff}$ emitted by petroleum (pet) and natural gas (ng) combustion, $f_{pet}$ and $f_{ng}$, are calculated from the values of $\delta_{ff}$:

$$\delta_{ff} = \delta_{pet} \times f_{pet/ff} + \delta_{ng} \times \left(1 - f_{pet/ff}\right) \tag{5}$$

$$f_{pet/ff} = \frac{\delta_{ff} - \delta_{ng}}{\delta_{pet} - \delta_{ng}} \tag{6}$$

We use values of $-25.5 \pm 0.5$ ‰ for $\delta_{pet}$ and $-40.2 \pm 0.5$ ‰ for $\delta_{ng}$ (Newman et al. 2008); $f_{pet} = f_{ff} \times f_{pet/ff}$, and $f_{ng} = f_{ff} \times f_{ng/ff}$, where $f_{ng/ff} = 1 - f_{pet/ff}$. We use temporally constant $\delta^{13}C$ signatures for petroleum, natural gas and biogenic sources (and sinks), although with additional processed-based information, this assumption could be relaxed in the future. Note that although pet, ng, and bio signatures are fixed, both $\delta_{src}$ and $\delta_{ff}$ vary with time, meaning that $f_{bio}$, $f_{pet}$ and $f_{ng}$ all vary at the frequency of the air sampling.

## 2.3 Partitioning CO$_2$ signals using CO and $\delta^{13}$CO2 measurements

Although we can determine $f_{bio}$, $f_{pet}$ and $f_{ng}$ at the frequency of discrete flask sampling events using the method described in Section 2.2, here we describe how comparable CO$_2$xs fractions can in theory be determined at high frequency using continuous measurements of CO and $\delta^{13}$CO$_2$. To evaluate the method, we compute the relative contributions of biogenic, petroleum and natural gas sources to CO$_2$xs using flask air CO and $\delta^{13}$CO$_2$ measurements and compare these to values obtained using the $\Delta^{14}$CO$_2$-guided approach for the same samples by applying the following system of equations:

$$R_{src} = R_{bio} \times f_{bio} + R_{pet} \times f_{pet} + R_{ng} \times f_{ng} \tag{7}$$

$$\delta_{src} = \delta_{bio} \times f_{bio} + \delta_{pet} \times f_{pet} + \delta_{ng} \times f_{ng} \tag{8}$$

$$1 = f_{bio} + f_{pet} + f_{ng} \tag{9}$$

$R_{src}$ represents the CO/CO$_2$ ratio of the total source, which is the observed COxs/CO$_2$xs ratio, and we use $R$ to refer to the CO/CO2 emission ratios of individual CO$_2$xs components (bio, pet, and ng). For now, we assume that $R$ of each source are constant over a year-long period and over the





greater LA region (discussed in Section 3.2); especially with high frequency CO and $\delta^{13}CO_2$ measurements, this assumption could easily be relaxed (discussed in Section 3.3).

$R$ values and $\delta^{13}C$ signatures for bio, pet, and ng are needed to solve Eqs. 7-9. $\delta^{13}C$ signatures are specified in section 2.2; $R$ values are obtained via multiple linear regression of Eq. 7 using observed $R_{src}$ and $f$ values determined using $\Delta^{14}C$ and $\delta^{13}C$ of $CO_2$ measurements as described in Section 2.2. Then we solve Eqs. 7-9 for new $f$ values, $f'$. This new $CO_2$xs partitioning (i.e., $f'_{bio}$, $f'_{pet}$, $f'_{ng}$) based on CO and $\delta^{13}CO_2$ observations is used to calculate new values of $C_{ff}$ and $C_{bio}$ (i.e., $C'_{ff}$ and $C'_{bio}$).

## 210 3    Results and Discussion

### 3.1 Contribution of biogenic, petroleum and natural gas sources in CO₂ excess

We calculated the fractional contribution of petroleum, natural gas, and biospheric fluxes to total $CO_2$xs each month from April 2015 to March 2016 using $\Delta^{14}CO_2$ and $\delta^{13}CO_2$ observations recorded at FUL, USC and GRA. The results are given in Table S1 and presented in Figure 3.

Figure 4 presents the results in terms of the relative $CO_2$xs contribution from each source at each site. We observe seasonal variation in $CO_2$xs from each source. Fossil fuel is the dominant $CO_2$ emissions source at each site which agrees with the findings of Newman et al. (2016) and Miller et al. (2020). Annually averaged across all three sites, biogenic emissions account for 6 % of $CO_2$xs. Biogenic emissions are larger and positive in winter and smaller and negative in summer,

indicating winter respiration and uptake in summertime, generally consistent with the results of Miller et al. (2020). Note that in this study, we do not partition $C_{bio}$ into an urban biosphere component and other components related to the oxidation of biogenic carbon including ethanol added to gasoline, and human and other animal food and waste (which can only be positive and are unlikely to vary much seasonally). If, as in Miller et al. (2020), we accounted for the always

positive ethanol, food, and waste signals, we would likely observe similarly large seasonal drawdown associated with urban vegetation.

We also observe spatial differences: The USC site exhibits a smaller contribution of the biosphere (3 % of annual average $CO_2$ excess) compared to FUL and GRA (9 % and 5 % of total $CO_2$ excess, respectively). However, these modest annual average biospheric contributions mask significant

seasonal activity. On a monthly basis, maximum positive biogenic contribution is observed in November at 25, 26, and 22 % at USC, FUL, and GRA (percentage of total $CO_2$ excess, respectively). And the maximum negative contribution, driven by net photosynthesis, is observed in July with values of -22, -13, and -12 % at USC, FUL, and GRA (percentage of total $CO_2$ excess, respectively).

Network average $C_{ff}$ is 11.0 ± 14.5 ppm in winter (November-February; median and standard deviation) and 12.2 ± 6.6 ppm in summer (May-August). No significant difference is observed in winter and summer $C_{ff}$. This corresponds to the seasonality in Hestia-LA emissions, which indicates $C_{ff}$ inputs are only 3 % higher in winter. High variability observed in wintertime $C_{ff}$ agrees with Miller et al. (2020) which is likely caused by increased temperature inversion trapping

as the cold ground surface in winter cools the air layer right above the ground. While bottom-up fossil fuel emissions reveal little seasonality, the top-down seasonality of petroleum and natural gas (Fig. 4), which as fractions of $C_{ff}$ should be largely independent of mixing, are clearly evident. The proportion of natural gas in fossil fuel signals are 40 % and 36 % in summer and 34 % and 30% in winter at FUL and USC (Fig. 3). The increase in the natural gas contribution observed in

summer can be explained by the increase in natural gas generated electricity in LA power plants to provide for air conditioning in summer (Newman et al. 2016; He et al. 2019). GRA, located northwest of USC by ~35 km without an electricity generation facility nearby, shows the opposite pattern (24 % in summer and 40 % in winter). This suggests the local influence of increased natural gas usage for heating in the winter.

**3.2 *R* values of biogenic, petroleum and natural gas sources**

Monthly, site-based $R_{src}$ varies between $5.5 - 11.4$ ppb ppm$^{-1}$ (Fig. 5), with a mean and standard deviation of 8.2 ± 1.6 ppb/ppm (relative s.d. = 19 %). Greater variability is seen in $R_{ff}$ (lower panel): mean and s.d. of 9.6 ± 2.1 ppm (relative s.d. = 22 %). To understand and predict the variation in $R_{ff}$, we further divide the fossil fuel emissions into petroleum and natural gas

emissions. Applying the calculated $f$ values from $\Delta^{14}CO_2$ and $\delta^{13}CO_2$ observations (Section 2.2., Fig. 3), we solve Eq. 7 for each source's $CO/CO_2$ emission ratio, $R$ (Table 1). Note that we exclude negative flask-based values of COxs (and corresponding $R_{src}$ values) and CO2ff (and corresponding $f_{ff}$ values) as non-physical. Likewise, positive $\delta_{src}$ values and $f_{pet/ff}$ values (and





corresponding $f_{pet}$ and $f_{ng}$ values) outside the range of 0-1 are also excluded. A bootstrapping

method is used to calculate the mean and uncertainty of possible $CO/CO_2$ ratios. The $CO/CO_2$

ratios of petroleum ($R_{pet}$) and natural gas ($R_{ng}$) combustion emissions are $12.2 \pm 0.6$ ppb ppm⁻¹

and $2.3 \pm 1.2$ ppb ppm⁻¹, respectively. As discussed above, the proportion of natural gas in fossil

fuel emissions is bigger in summer resulting in smaller $R_{ff}$ in summer at FUL and USC. We find

the value of $1.8 \pm 0.8$ ppb ppm⁻¹ for $R_{bio}$ which is non-zero because biofuel (mainly corn-based

ethanol) in the gasoline in California with large $CO/CO_2$ ratio signal is included in the biogenic

sources while respiratory $CO/CO_2$ ratios approach 0. A larger contribution of the biosphere with a

low $CO/CO_2$ ratio in winter offsets the large $R_{ff}$ lowering the variability in $R_{src}$ at each site.

We compare our model-determined $CO/CO_2$ ratios of each source (Table 2) to bottom-up

inventory-based estimates (Table 1). $CO/CO_2$ ratios of each source constrained from our model

and observational data approach agree well with the bottom-up inventory-based estimates. Sources

contributing a high percentage of $CO_2$ emissions strongly influence the total $CO/CO_2$ ratio. The

$CO/CO_2$ ratio of petroleum combustion is greatly affected by on-road emissions and industrial

emissions (contributing 28 % and 60 % of total petroleum $CO_2$ emissions). Natural gas is mostly

dominated by non-mobile emissions (Electricity production, residential, commercial, and

275 industrial, sequentially) resulting in low $CO/CO_2$ ratio.

### 3.3 Estimation of $CO_2ff$ based on CO and ¹³$CO_2$ observations

Table 2 shows the $CO/CO_2$ ratio and δ¹³C signature of each source. The combination of the $R$ and

$δ$ signals reveal a distinct pattern for each source: the biosphere has low near-zero $R$, petroleum

has high $R$, and natural gas has low $R$. Petroleum and biosphere $CO_2$ have similar $δ$ values,

whereas natural gas has a very low $δ$. By substituting these values into Eqs. 7-9, $f'$ values are

calculated, and then we calculate $C'_{ff}$ by multiplying the sum of $f'_{pet}$ and $f'_{ng}$ by $CO_2xs$

measured every few days. We compare $f'_{ff}$ and $C'_{ff}$ to $f_{ff}$ and $C_{ff}$ (determined using ¹⁴C

observations) in Fig. 6. Assessment for each source is shown in Figure S2 and S3. The $R^2$ values

are 0.63 and 0.90 for $f'_{ff}$ and $C'_{ff}$, respectively.

If $R$ values are allowed to vary in time, it is likely to improve the precision of the method. We

calculate the uncertainty in $C'_{ff}$ for varying temporal resolutions of $R$ (black solid line in Fig. 7).

We find that the uncertainty increases when the size of the window increases from 1-week to 10-



week; in other words, allowing temporal variation in $R$ improves the precision of the method. However, the uncertainty slightly decreases beyond the 10-week window. This is likely caused by the reduction in the error of $R$ values (not shown) as the number of observations used to find $R$ (by solving Eq. 7) increases. In summary, the ideal flask sampling frequency for this method would be higher than every 2 weeks. In cases where this is impossible, it is better to assume constant $R$ values.

The uncertainty in $C'_{ff}$ estimated using the CO-based method is also shown in Figure 7 (black dashed line). The CO-based method also provides improved precision of the method when flask sampling is available at higher frequency. However, the CO and $\delta^{13}CO_2$-based method shows greater confidence than the CO-based method for the whole range of adjusted temporal resolution in $R$. When using constant $R$ values (temporal resolution of 50 weeks), the uncertainty is 3.2 ppm (the 1σ standard deviation of differences between $C_{ff}$ and $C'_{ff}$) for the CO and $\delta^{13}CO_2$-based method, while it is 4.8 ppm for CO-based method. This improvement is likely associated with the additional information provided by $\delta^{13}CO_2$ that constrains the effective $R_{ff}$ and further separates fossil fuel sources into sub-categories (petroleum and natural gas sources).

## 4 Conclusions

We present a CO and $\delta^{13}CO_2$-based method to estimate $CO_2ff$ which is based on flask-based $\Delta^{14}CO_2$ measurements. We have applied the method to measurements from flask samples collected in the LA basin, every few days in the afternoon for more than one year (2015-16). The proposed method was assessed by comparing it to a more traditional $\Delta^{14}CO_2$-based method. CO and $\delta^{13}CO_2$ approach can be applied to continuous measurements of $CO_2$, CO and $\delta^{13}CO_2$ which can provide $CO_2ff$ estimates at higher temporal resolution and with greater accuracy than previously applied CO-based methods.

We have analyzed three locations in the Los Angeles megacity, partitioning observed $CO_2$ enhancements ($CO_2xs$) into biogenic, petroleum and natural gas sources. We observed a substantial biogenic signal that varies from -14% to +25% of $CO_2xs$ over the course of the year, with positive contributions in winter and negative contributions in summer due to net respiration and net photosynthesis, respectively. Furthermore, partitioning $CO_2ff$ into petroleum and natural gas combustion fractions revealed that natural gas combustion has the largest contribution in



summer, potentially due to an increase in electricity generation at LA power plants for air conditioning.

**Data availability.** The data that support the findings of this study are available from JBM (john.b.miller@noaa.gov) upon request.

**Author contributions.** JK designed and executed the study. JBM, SJL, and SEM provided the data. JK prepared the manuscript with contributions from all co-authors

**Competing interests.** The authors declare that they have no conflict of interest.

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

Methods, Measurement, and Modeling." *Radiocarbon* 55 (3).
https://doi.org/10.1017/s0033822200048414.

Levin, Ingeborg, and Ute Karstens. 2007. "Inferring High-Resolution Fossil Fuel CO2 Records
at Continental Sites from Combined 14CO2 and CO Observations." *Tellus, Series B:
Chemical and Physical Meteorology* 59 (2): 245–50. https://doi.org/10.1111/j.1600-
0889.2006.00244.x.

Lopez, Morgan, M. Schmidt, M. Delmotte, A. Colomb, V. Gros, C. Janssen, S. J. Lehman, et al.
2013. "CO, NOx and 13CO2 as Tracers for Fossil Fuel CO2: Results from a Pilot Study in
Paris during Winter 2010." *Atmospheric Chemistry and Physics* 13 (15): 7343–58.
https://doi.org/10.5194/acp-13-7343-2013.

McCartt, A. Daniel, and Jun Jiang. 2022. "Room-Temperature Optical Detection of 14CO2below
the Natural Abundance with Two-Color Cavity Ring-Down Spectroscopy." *ACS Sensors* 7
(11). https://doi.org/10.1021/acssensors.2c01253.

Mcdonald, Brian C., Zoe C. McBride, Elliot W. Martin, and Robert A. Harley. 2014. "High-
Resolution Mapping of Motor Vehicle Carbon Dioxide Emissions." *Journal of Geophysical
Research: Atmospheres*, no. May: 5283–98.
https://doi.org/10.1002/2013JD021219.Received.

Miller, John B., Scott J. Lehman, Stephen A Montzka, Colm Sweeney, Benjamin R Miller, Anna
Karion, Chad Wolak, et al. 2012. "Linking Emissions of Fossil Fuel CO2 and Other
Anthropogenic Trace Gases Using Atmospheric 14CO2." *Journal of Geophysical Research:
Atmospheres* 117. https://doi.org/10.1029/2011JD017048.

Miller, John B., Scott J. Lehman, Kristal R. Verhulst, Charles E. Miller, Riley M. Duren, Vineet
Yadav, Sally Newman, and Christopher D. Sloop. 2020. "Large and Seasonally Varying
Biospheric CO2 Fluxes in the Los Angeles Megacity Revealed by Atmospheric
Radiocarbon." *Proceedings of the National Academy of Sciences of the United States of
America* 117 (43): 26681–87. https://doi.org/10.1073/pnas.2005253117.





Newman, Sally, and S. Jeong. 2013. "Diurnal Tracking of Anthropogenic CO2 Emissions in the
         Los Angeles Basin Megacity during Spring 2010." *Atmospheric Chemistry and Physics*,
         4359–72. https://doi.org/10.5194/acp-13-4359-2013.

Newman, Sally, Xiaomei Xu, Hagit P. Affek, Edward Stolper, and Samuel Epstein. 2008.
         "Changes in Mixing Ratio and Isotopic Composition of CO2 in Urban Air from the Los
Angeles Basin, California, between 1972 and 2003." *Journal of Geophysical Research
         Atmospheres* 113 (23): 1–15. https://doi.org/10.1029/2008JD009999.

Newman, Sally, Xiaomei Xu, K. R. Gurney, Ying Kuang Hsu, King Fai Li, Xun Jiang, Ralph F.
         Keeling, et al. 2016. "Toward Consistency between Trends in Bottom-up CO2 Emissions
         and Top-down Atmospheric Measurements in the Los Angeles Megacity." *Atmospheric
Chemistry and Physics* 16 (6): 3843–63. https://doi.org/10.5194/acp-16-3843-2016.

Sargent, Maryann, Yanina Barrera, Thomas Nehrkorn, L. R. Hutyra, C. K. Gately, Taylor Jones,
         Kathryn McKain, et al. 2018. "Anthropogenic and Biogenic CO2 Fluxes in the Boston
         Urban Region." *Proceedings of the National Academy of Sciences* 115 (29): 7491–96.
         https://doi.org/10.1073/pnas.1803715115.

Sweeney, Colm, Anna Karion, Sonja Wolter, Timothy Newberger, Doug Guenther, Jack A
         Higgs, Arlyn Elyzabeth Andrews, et al. 2015. "Seasonal Climatology of CO2 across North
         America from Aircraft Measurements in the NOAA/ESRL Global Greenhouse Gas
         Reference Network." *Journal of Geophysical Research : Atmospheres*, 5155–90.
         https://doi.org/10.1002/2014JD022591.Received.

Turnbull, J. C., Anna Karion, Kenneth J. Davis, Thomas Lauvaux, Natasha L. Miles, Scott J.
         Richardson, Colm Sweeney, et al. 2019. "Synthesis of Urban CO2 Emission Estimates from
         Multiple Methods from the Indianapolis Flux Project (INFLUX)." *Environmental Science
         and Technology* 53 (1): 287–95. https://doi.org/10.1021/acs.est.8b05552.

Turnbull, J. C., Colm Sweeney, Anna Karion, Timothy Newberger, Scott J. Lehman, Pieter P.
Tans, Kenneth J. Davis, et al. 2015. "Toward Quantification and Source Sector
         Identification of Fossil Fuel CO2 Emissions from an Urban Area: Results from the
         INFLUX Experiment." *Journal of Geophysical Research* 120 (1): 292–312.
         https://doi.org/10.1002/2014JD022555.

Turner, Alexander J., Jinsol Kim, Helen Fitzmaurice, Catherine Newman, Kevin Worthington,
Katherine Chan, Paul Wooldridge, Philipp Köhler, Christian Frankenberg, and Ronald C.



Cohen. 2020. "Observed Impacts of COVID-19 on Urban CO2 Emissions." *Geophysical Research Letters*, 2–10.

Vardag, S. N., C. Gerbig, G. Janssens-Maenhout, and Ingeborg Levin. 2015. "Estimation of Continuous Anthropogenic CO2: Model-Based Evaluation of CO2, CO, Δ13C(CO2) and Δ14C(CO2) Tracer Methods." *Atmospheric Chemistry and Physics* 15 (22): 12705–29. https://doi.org/10.5194/acp-15-12705-2015.

Vaughn, B. H., John B. Miller, D. F. Ferretti, and J. W.C. White. 2004. "Stable Isotope Measurements of Atmospheric CO2 and CH4." In *Handbook of Stable Isotope Analytical Techniques*. https://doi.org/10.1016/B978-044451114-0/50016-8.

Vogel, Felix R., Matthias Frey, Johannes Staufer, Frank Hase, Grégoire Broquet, and Irène Xueref-remy. 2019. "XCO2 in an Emission Hot-Spot Region : The COCCON Paris Campaign 2015." *Atmospheric Chemistry and Physics* 19: 3271–85.

Vogel, Felix R., Samuel Hammer, Axel Steinhof, Bernd Kromer, and Ingeborg Levin. 2010. "Implication of Weekly and Diurnal 14C Calibration on Hourly Estimates of CO-Based Fossil Fuel CO2 at a Moderately Polluted Site in Southwestern Germany." *Tellus, Series B: Chemical and Physical Meteorology* 62 (5): 512–20. https://doi.org/10.1111/j.1600-0889.2010.00477.x.

Wu, Dien, John C. Lin, Henrique F. Duarte, Vineet Yadav, Nicholas C. Parazoo, Tomohiro Oda, and Eric A. Kort. 2021. "A Model for Urban Biogenic CO2 Fluxes: Solar-Induced Fluorescence for Modeling Urban Biogenic Fluxes (SMUrF V1)." *Geoscientific Model Development* 14 (6). https://doi.org/10.5194/gmd-14-3633-2021.

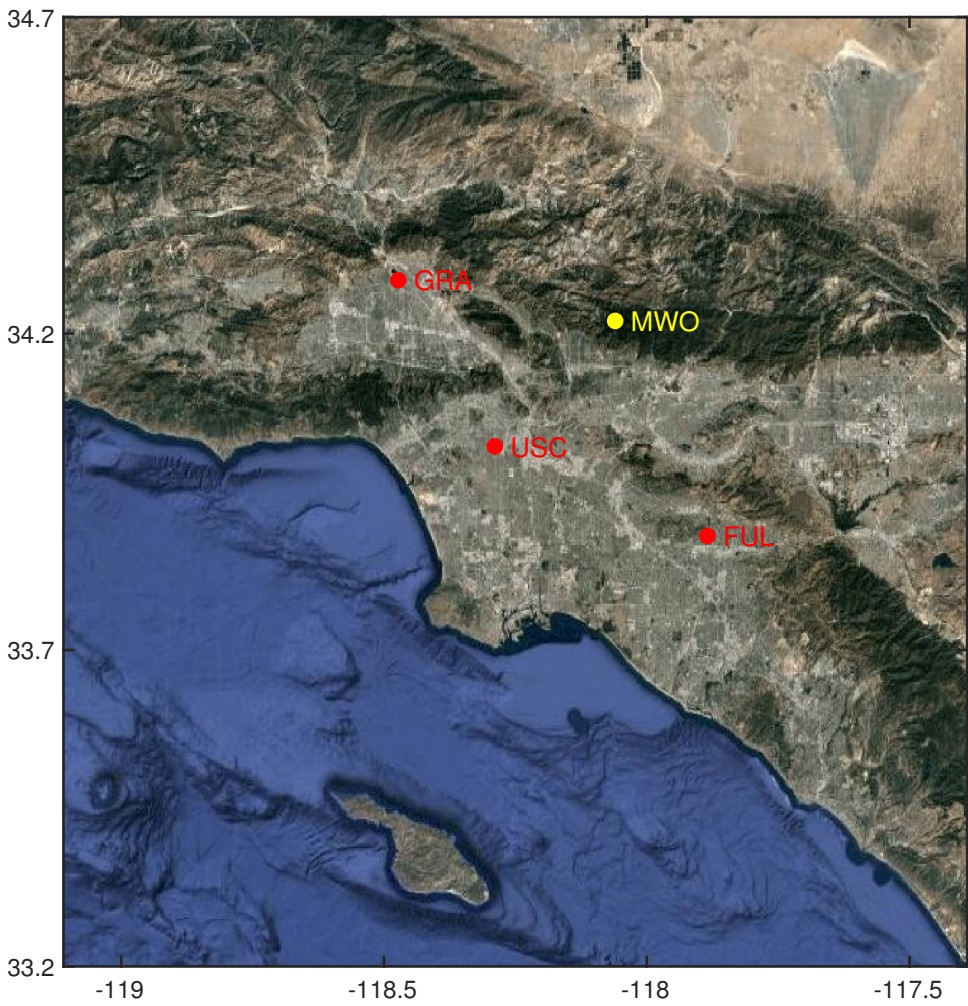

**Figure 1.** Map of the greater Los Angeles region. The three Los Angeles Megacity Carbon Project sites are marked in red and the Mount Wilson Observatory used to define background values are marked in yellow. Map data © Google Maps 2022.



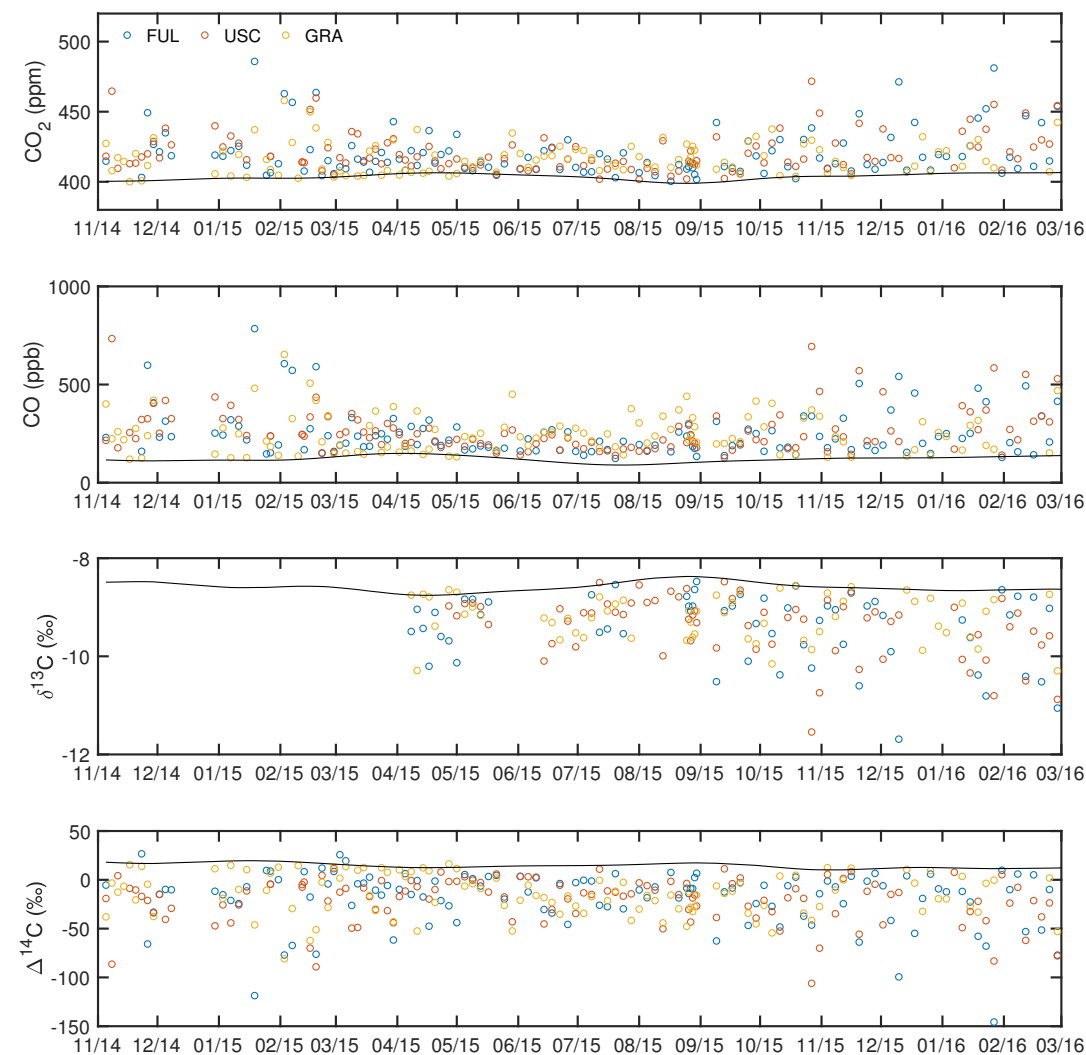

**Figure 2.** Timeseries of $CO_2$, CO, $\delta^{13}C$, and $\Delta^{14}C$. Black line represents background values.



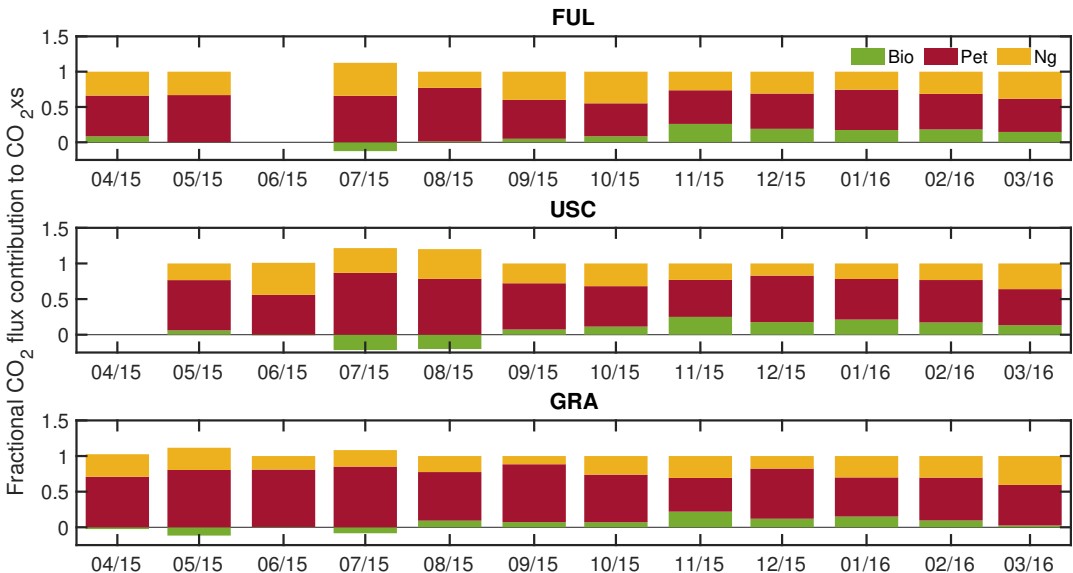

**Figure 3.** Monthly mean fractional contributions (f) of biosphere (green), petroleum (red), and natural gas (yellow) to $CO_2$xs at each site, as determined from $\Delta^{14}C$ and $\delta^{13}C$ observations (Section 2.2). The sum of the fractions is one in each month.

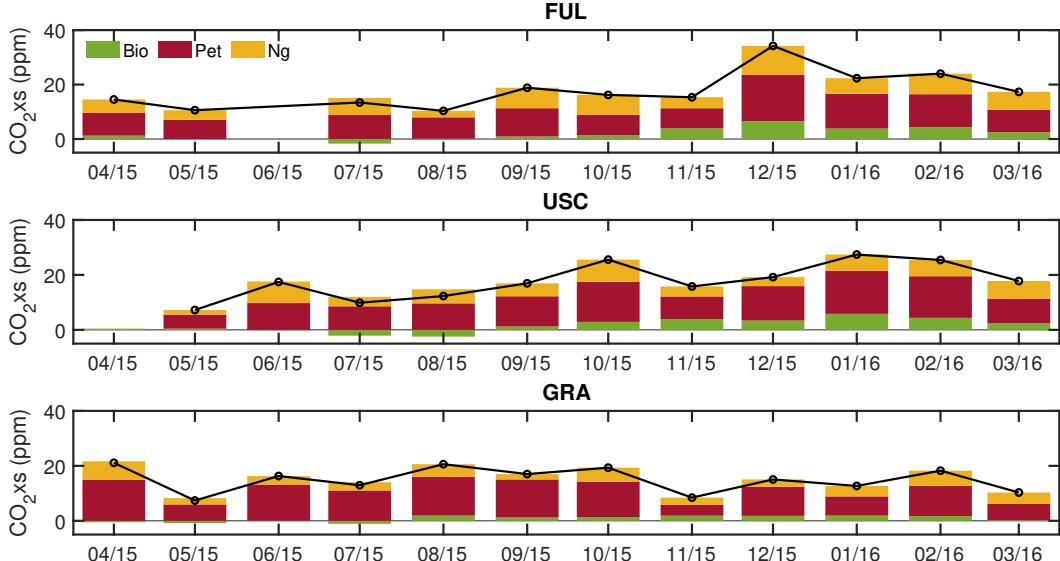

**Figure 4.** Monthly mean $CO_2xs$ partitioned into biosphere (green), petroleum (red), and natural gas (yellow) signals, as determined from $\Delta^{14}C$ and $\delta^{13}C$ observations, at each site. The black marker indicates $CO_2xs$.





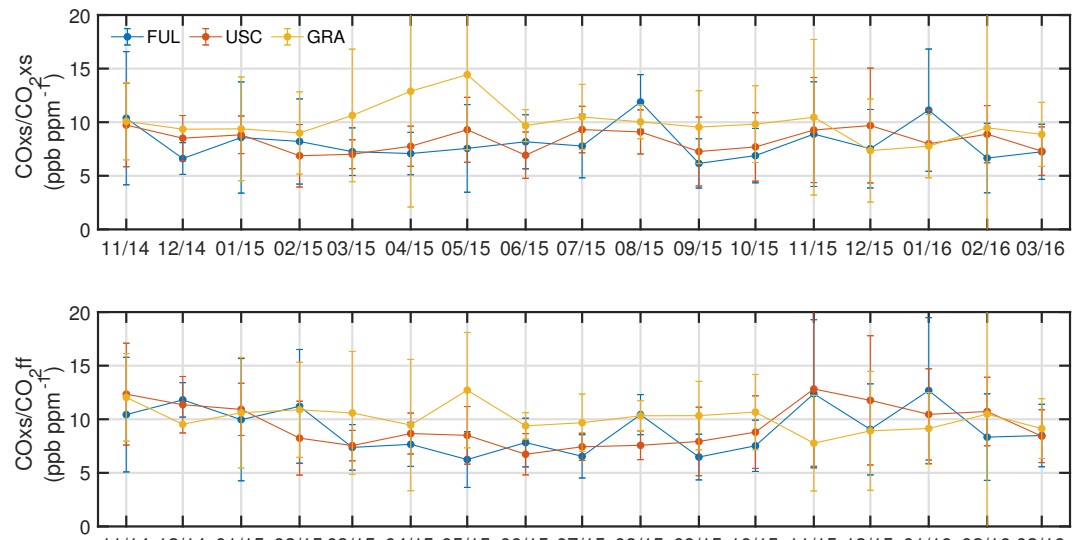

**Figure 5.** Monthly variations in COxs/CO$_2$xs (Rsrc) and COxs/CO$_2$ff (Rff) at each site. COxs/CO$_2$ff is calculated using $^{14}$C observations.



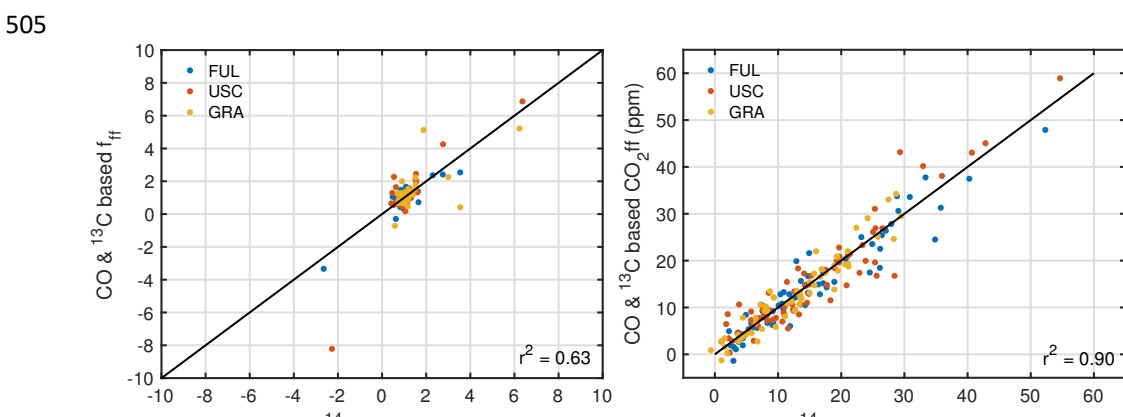

**Figure 6.** Comparison of $f_{ff}$ and $f'_{ff}$ (left) and $C_{ff}$ and $C'_{ff}$ (right). Black lines represent 1:1 relationships and different colors indicate different sites.



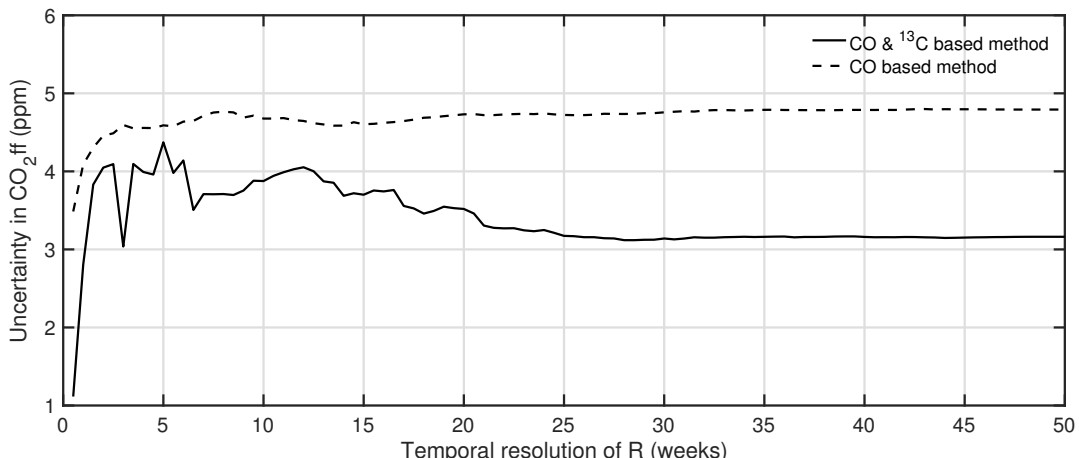

**Figure 7.** Uncertainty in $C'_{ff}$ for varying temporal resolution of $R$ (N weeks). $R$ is determined for each data point solving Eq. 7 using CO, $^{13}CO_2$ and $^{14}CO_2$ observations within a moving window of 2N weeks. For CO-based method, $R_{ff}$ is smoothed using a 2N weeks moving window.





**Table 1.** Bottom-up $CO_2$ emission, CO emission, and R ($CO/CO_2$ ratio) estimates for each
source sector and fuel type for LA basin based on the Vulcan 3.0 and the U.S. Environmental
Protection Agency (EPA) National Emission Inventory for 2011 (NEI 2011) product. NEI 2011
is scaled by the emissions with fuel consumption dataset from the U.S. Energy Information
Administration (EIA) State Energy Data System (SEDS) to estimate 2015 CO emissions.

| | ------------Petroleum------------- | | | -----------Natural Gas---------- | | |
| | $CO_2$ (MtC) | CO (MtC) | R = $CO/CO_2$ (ppb ppm$^{-1}$) | $CO_2$ (MtC) | CO (MtC) | R = $CO/CO_2$ (ppb ppm$^{-1}$) |
|---|---|---|---|---|---|---|
| Residential | 0.06 | <0.001 | 0.07 | 2.79 | 0.001 | 1.21 |
| Commercial | 0.67 | <0.001 | 0.34 | 1.79 | 0.002 | 2.54 |
| Industrial | 9.79 | <0.001 | 0.06 | 1.59 | 0.002 | 2.99 |
| Electricity Production | 0.37 | <0.001 | 0.04 | 5.08 | 0.002 | 0.74 |
| On-road | 20.97 | 0.296 | 31.79 | 0 | 0 | |
| Non-road | 1.45 | 0.139 | 224.11 | 0.19 | 0.012 | 152.41 |
| Airport | 0.89 | 0.008 | 21.63 | 0 | 0 | |
| Rail | 0.47 | 0.002 | 11.94 | 0 | 0 | |
| CMV | 0.48 | <0.001 | 3.26 | 0 | 0 | |
| **Total** | **35.16** | **0.437** | **12.42** | **11.44** | **0.019** | **1.68** |



**Table 2.** CO/CO$_2$ ratios ($R$) and $\delta^{13}$C signatures used to determine relative contribution of biogenic, petroleum and natural gas sources.

|  | Biosphere | Petroleum | Natural Gas |
|---|---|---|---|
| Bottom-up approach $R$ (ppb ppm$^{-1}$)[a] |  | 12.4 | 1.7 |
| Top-down approach $R$ (ppb ppm$^{-1}$)[b] | 1.8 ± 0.8 | 12.2 ± 0.6 | 2.3 ± 1.2 |
| $\delta^{13}$C (‰)[c] | -26.6 ± 0.5 | -25.5 ± 0.5 | -40.2 ± 0.5 |

[a]$R$ calculated from Table 1. These values are not used for CO$_2$xs partitioning and are for reference only.
[b]$R$ calculated from CO, $\delta^{13}$CO$_2$ and $\Delta^{14}$CO$_2$ flask observations. These values are used in this study.
[c]$\delta^{13}$C from previous studies (Newman et al., 2008)