# Peer review of "Quantification of fossil fuel CO2 from combined CO, $\delta^{13}$ CO2 and $\Delta^{14}$ CO2 observations"

_EGUsphere, 2023_

## Author Comment (AC1)

**Response to Referee Comments:**

We thank the referees for their detailed comments which have been a great help to improve our manuscript.
* * *
**Referee #1 Comments: (Referee comments in italics)**

*Kim et al. present method for source apportionment of urban CO2 sources in the Greater Los Angeles (LA) area based on atmospheric observations of CO, CO2, 13CO2 and 14CO2. They use the difference of each species compared to background observations at MWO to infer contributions from biospheric CO2 and CO2 from fossil fuel burning further separated into CO2 from gasoline and natural gas combustion.*
*The authors find significant contributions from natural sources in the LA area as well as seasonal changes in the contribution from natural gas and gasoline combustion. Lastly, the study briefly assesses the uncertainty of different source apportionment methods.*
*Overall, the paper is straightforward, well-written, easy to follow and clear in its methods, results and discussion. The method is an advancement of previously published source apportionment methods and will contribute to the continuously growing field of research focussing on greenhouse gas emissions from urban areas, which will surely be of interest to the readers of ACP. Before publication, however, the issue of the petroleum source signature should be addressed as well as some other minor and technical comments.*

**We thank the reviewer for the positive comments.**

General comments:

*The choice of d13Cpet as 25.5+/0.5permil is unclear. This is not the exact number suggested for CO2 from car exhaust by Newman et al. 2008 (see comment L 183). Furthermore, it would be prudent to validate the value of d13Cpet given in Newman et al. as the sources of crude oil being refined in California has changed significantly over the last 20 years. Alternatively, at least an assessment of the impact a different source signature would have on the results should be added.*

*L183: Why was a value of -25.5+/-0.5 permil chosen for $d^{13}C_{pet}$?*
*Table A1 in Newman et al. 2008 lists 2002-2003 petroleum composite as -25.8+/-0.5permil, but automobile exhaust was reported at -26.0+/-0.2permil. Furthermore, this signature strongly depends on the source of the crude oil (local less depleted vs important more depleted crude). The local California field production has fallen from 700k bbl/day in 2002-2003 to ca. 300k bbl/day 2022-2023 according to https://www.eia.gov/, thus the gasoline d13C has likely changes in the last 20 years. (see also attached figure from https://www.energy.ca.gov/)*

**A value of -25.5+/-0.5 permil is based on measurements in 2014 reported in Newman et al., (2016). This value is similar to the estimated d13Cpet value (-25.6 permil) using the source of the crude oil (*https://www.energy.ca.gov/*) and d13C for each source reported in Newman et al., (2008). We will correct the citation on the manuscript.**

Specific/technical comments:

*L40: bottom-up emission inventories/models do not solely rely on consumption data as suggested here but can also use activity data to estimate emissions. For on-road vehicle emissions, vehicle kilometers/miles travelled (VKT/VMT) is a common proxy activity used.*

**We will correct L40 as follows:**

**"Current understanding of anthropogenic $CO_2$ emissions mainly derives from methods that estimate aggregate emissions in a domain using economic statistics such as total fuel sales or activity data such as total distance traveled for on-road vehicle emissions."**

*L46: there is important work preceding Heimburger et al. e.g. : Mays et al. 2009 https://doi.org/10.1021/es901326b; Cambaliza et al. 2014, https://doi.org/10.5194/acp-14-9029-2014*

**We will add the studies to the reference list.**

*L48: Preceding work by Breon et al. 2015 https://doi.org/10.5194/acp-15-1707-2015; is not mentioned*

**We will add the study to the reference list.**

*L57: How can Hardiman et al. 2017 be cited as evidence that recent improvements in biosphere models like Gourdji et al. 2022 are insufficient, given that Hardiman et al. was published 5 years before the improvements by Gourdji et al. were?*

**We will add a following study to the reference list:**

**Winbourne, J. B., I. A. Smith, H. Stoynova, C. Kohler, C. K. Gately, B. A. Logan, J. Reblin, A. Reinmann, D. W. Allen, and L. R. Hutyra. 2022. "Quantification of Urban Forest and Grassland Carbon Fluxes Using Field Measurements and a Satellite-Based Model in Washington DC/Baltimore Area." *Journal of Geophysical Research: Biogeosciences* 127 (1). https://doi.org/10.1029/2021JG006568.**

*L231: please add % after every number that is expressed in percent. (Like done in line 243)*

**We will make those edits to the manuscript.**

*L250: ppb ppm-1 is not exactly the unit here as you comparing moles of CO per moles of air with moles of CO2 with moles of air. Suggestion to either change unit to ppbCO ppmCO2$^{-1}$, explain your shorthand briefly or change the title of the section to "CO:CO2 ratio (R) values of…" to avoid confusion for the reader.*

**We will change the title of the section to:**

**"CO:CO₂ emission ratio (R) values of biogenic, petroleum and natural gas sources"**

L465: Figure 1. The labels for GRA, USC and Ful have little contrast, suggestion to change color.

**We have changed the color for GRA, USC and FUL as suggest by reviewer.**

L500: The labels on the ordinate overlap with the units and the abscissa lacks the label (date in xx/xx)

**We have updated the Figure and the caption of Figure 5 as suggested by the reviewer.**
* * *
**Referee #2 Comments: (Referee comments in italics)**

*This paper describes the partitioning of $CO_2$ enhancements over Los Angeles (relative to incoming air) into biogenic, petroleum and natural gas components, using a combination of three different tracers – CO, $\partial^{13}CO_2$ and $\Delta^{14}CO_2$. This is a very nice study presenting a clever new way to partition emissions, and is entirely suitable for publication in Atmospheric Chemistry and Physics. I recommend some fairly minor changes prior to publication.*

**We thank the referee for their positive and constructive comments.**

Specific comments:
*Please check for subscripts/superscripts throughout, and there are a few grammatical errors that should be readily fixable with a careful reading or two.*

**We will correct subscripts/superscripts throughout the text.**

*Lines 19-21. It would be helpful to give the typical magnitude of the $CO_2xs$ signal to put the uncertainty in $CO_2ff$ into context. From Figure 4, it looks like the typical magnitude is ~20 ppm so 3.2-4.8 ppm is a relatively large uncertainty.*

**We have updated the text as follow as suggested by the reviewer:**

**"Relative to calculating $CO_2ff$ (12.4 ± 10.9 ppm; median and standard deviation) directly from $\Delta^{14}CO_2$, we find that the uncertainty in $CO_2ff$ estimated from the CO and $\delta^{13}CO_2$-based method averages 3.2 ppm which is significantly less than the CO-based method which has an average uncertainty of 4.8 ppm."**

*Lines 38-39. The authors may want to refer to efforts such as the $IG^3IS$ Urban Guidelines here, and perhaps reference other papers that have demonstrated the need for monitoring systems. Turnbull JC, DeCola PL, Mueller K, Vogel F, Agusti-Panareda A, Ahn D, Baidar S, Bovensmann H, Brewer A, Brunner D et al. 2022. $IG^3IS$ Urban Greenhouse Gas Emission*

*Observation and Monitoring Good Research Practice Guidelines - WMO GAW IG³IS Report 275, 2021. Geneva: World Meteorological Organisation.*

**We will add the report to the reference list.**

*Lines 39-59. The references given in this section are almost exclusively examples from the US. Suggest adding some examples from other regions of the world.*

**We will add following studies to the reference list:**

**Bréon, F M, G Broquet, V Puygrenier, F Chevallier, M Ramonet, E Dieudonné, Morgan Lopez, U M R Cea-cnrs-uvsq, and Gif Yvette. 2015. "An Attempt at Estimating Paris Area CO2 Emissions from Atmospheric Concentration Measurements." *Atmospheric Chemistry and Physics*, 1707–24. https://doi.org/10.5194/acp-15-1707-2015.**

**Staufer, Johannes, Grégoire Broquet, F M Bréon, Vincent Puygrenier, Frédéric Chevallier, Irène Xueref-remy, Elsa Dieudonné, et al. 2016. "The First 1-Year-Long Estimate of the Paris Region Fossil Fuel CO2 Emissions Based on Atmospheric Inversion." *Atmospheric Chemistry and Physics*, 14703–26. https://doi.org/10.5194/acp-16-14703-2016.**

**Super, Ingrid, Stijn N.C. Dellaert, Antoon J.H. Visschedijk, and Hugo A.C.Denier Van Der Gon. 2020. "Uncertainty Analysis of a European High-Resolution Emission Inventory of CO2 and CO to Support Inverse Modelling and Network Design." *Atmospheric Chemistry and Physics* 20 (3). https://doi.org/10.5194/acp-20-1795-2020.**

*Lines 53-55. Can you clarify this statement – I think I understand that these previous studies have assumed that the biogenic emissions are "known" and the inversion has therefore solved only for fossil fuel $CO_2$?*

**We agree with the reviewer. We will rephrase the text as follows:**

**"Previous top-down studies used biosphere models to estimate biogenic fluxes and then focused on determining the balance of emissions attributable to fossil fuel combustion assuming that the biogenic emissions are known (Sargent et al. 2018; Turner et al. 2020; Lauvaux et al. 2020)."**

*Line 63. Reference Stuiver and Polach 1977.*
*Stuiver M, Polach HA. 1977. Discussion: Reporting of ¹⁴C data. Radiocarbon. 19(3):355-363.*

**We will add the study to the reference list.**

*Lines 74-76.  There are several examples of using $R_{ff}$ for aircraft campaigns, where time variability in $R_{ff}$ is not a concern, such as:*
*Graven HD, Stephens BB, Guilderson TP, Campos TL, Schimel DS, Campbell JE, Keeling RF. 2009. Vertical profiles of biospheric and fossil fuel-derived CO2 and fossil fuel $CO_2$ : CO ratios from airborne measurements of $\Delta^{14}C$, $CO_2$ and CO above Colorado, USA. Tellus B. 61(3):536-546.*
*Turnbull JC, Karion A, Fischer ML, Faloona I, Guilderson T, Lehman SJ, Miller BR, Miller JB, Montzka S, Sherwood T et al. 2011. Assessment of fossil fuel carbon dioxide and other anthropogenic trace gas emissions from airborne measurements over Sacramento, California in spring 2009. Atmospheric Chemistry and Physics. 11(2):705-721.*

**We have added the following text as suggested by the reviewer:**

**"A few studies have applied this method to estimate fossil fuel emissions for a moment in time during an airborne measurement campaign (Graven et al. 2009; Turnbull et al. 2011)."**

*Lines 79-80.  There is also a potential CO source from oxidation of VOCs, particularly in summer.  See for example:*
*Vimont IJ, Turnbull JC, Petrenko VV, Place PF, Sweeney C, Miles N, Richardson S, Vaughn BH, White JWC. 2019. An improved estimate for the $^{13}C$ and $^{18}O$ signatures of carbon monoxide produced from atmospheric oxidation of volatile organic compounds. Atmospheric Chemistry and Physics. 19(13):8547-8562.*

**We will add the following text at the end of L80:**

**"Additionally, CO produced from oxidation of volatile organic compounds (VOCs) can have an effect (Vimont et al. 2019)."**

*Lines 81-82.  Dividing $CO_2ff$ into high and low CO sources has also been done for urban inversions, and it has worked quite well.  See:*
*Lauvaux T, Gurney KR, Miles NL, Davis KJ, Richardson SJ, Deng A, Nathan BJ, Oda T, Wang J, Hutyra LR et al. 2020. Policy-Relevant Assessment of Urban $CO_2$ Emissions. Environmental Science & Technology. 54(16):10237–10245.*

**This paragraph is reviewing the study that combines CO and $^{13}CO_2$. We will add the study to the reference list in L73.**

*Line 155.  Reference Craig 1957 or other suitable paper.*
*Craig H. 1957. Isotopic standards for carbon and oxygen and correction factors for mass-spectrometric analysis of carbon dioxide. Geochimica Et Cosmochimica Acta. 12:133-149.*

**We will add the study to the reference list.**

*Lines 155-158. I would expect that by calculating $\partial_{src}$ on a sample-by-sample basis, samples for which CO₂xs is small will have very large uncertainties. Is that the case, and how do you deal with that?*

**We do exclude samples with large uncertainties in $\partial_{src}$. When calculated $f_{pet/ff}$ (eq. 6) show values outside the range of 0 and 1, we exclude those samples based on $f_{pet/ff}$. We will add the following text at the end of L187:**

**"Samples with calculated $f_{pet/ff}$ values outside the range of 0 and 1, corresponding to small CO2xs and large uncertainty in $\delta_{src}$, are excluded from the analysis."**

*Line 176. Please add a sentence summarizing the result of the test of using a different $\partial bio$ value. Currently there is only a figure in the supplement, but no explanation in the main text.*

**We will add the following text in L176.**

**"When we change $\delta_{bio}$ from $-26.6$ ‰ to $-20$ ‰, $f_{pet}$ decreases by 0.04 and $f_{ng}$ increases by 0.05 which is smaller than the median uncertainty in $f_{pet}$ and $f_{ng}$ which is 0.17 and 0.16, respectively."**

*Line 182. Does the $\partial pet$ value include all petroleum sources? Are there differences in $\partial^{13}C$ for diesel, gasoline and other products? What is the $\partial^{13}C$ of biofuel additives? Are they large enough to matter? Have they changed through time?*

**The reviewer raises a good question about possible variation in $\partial pet$ value. There will certainly be changes in $\partial pet$ value based on a range of factors but it is very difficult to determine if the overall average value we use has changed within a year.**

**$\partial pet$ value includes all petroleum sources. $\partial pet$ value would vary depending on the source of crude oil (Newman et al., 2008). A value of -25.5+/-0.5 permil that we use in this study is based on measurements in 2014 reported in Newman et al., (2016). This value is similar to the estimated d13Cpet value (-25.6 permil) using the source of the crude oil (*https://www.energy.ca.gov/*) and d13C for each source reported in Newman et al., (2008).**

*Line 221-223. How exactly are biofuels (particularly ethanol in gasoline) accounted for in this multi-tracer analysis? This could be complicated, because $^{14}C$ will see biofuels as a biogenic source, but CO will see biofuels as a petroleum source since they will have a high R value. Please explain more carefully in the text.*

**We agree with the reviewer that separating biofuel signals is important and challenging. While we are also interested in quantifying the relative contribution of biofuel, separate from urban ecosystems, providing a thorough answer is beyond the aim of this study introducing a novel method of partitioning biogenic, petroleum, and natural gas sources and assessing the uncertainty. As R values are constrained using relative contribution calculated using $\partial^{13}CO_2$ and $\Delta^{14}CO_2$, biofuel signals are included as a biogenic source which leads to non-zero $R_{bio}$ value in Table 2 which is later discussed in section 3.2 (see L264).**

*Lines 241-249.  It would be useful to include a direct comparison of the relative contributions of petroleum and natural gas between the observations and Hestia.  Do they agree?*

**Contributions of petroleum relative to fossil fuel ($f_{pet/ff}$) show annual average of 67 % from the observations which is lower than the estimation of 75 % in Hestia. We will add the following text in L240:**

**"While Hestia-LA estimated relative contribution of petroleum and natural gas to fossil fuel emissions as 75 % and 25 %, we observe lower contribution of petroleum, 67 %, and larger contribution of natural gas, 33 %."**

*Line 256.  Table 2, not Table 1?*

**Apologies. We will correct the text.**

*Lines 268-270 and Table 1.  The summed $R_{pet}$ and $R_{ng}$ look reasonable, but the R values for each sector in Table 1 don't seem to make sense and can't add up to the summed value given.  Onroad, for example, shows R of 31, but by my calculation should be 14.*

**Apologies. We will fix the R values for each sector in Table 1. Only R values for Total is correct.**

*Also, did you consider using the CARB CO inventory?  In general, CARB does a much better job for CO than the NEI, but it may be that the NEI has simply adopted the CARB CO values for California.*

**We did consider using the CARB CO inventory, but decided to use NEI that categorized the emission sector simpler than the CARB CO inventory so as not to introduce errors by incorrect grouping of detailed sub-categories. Further, comparing R values in Table 2 to Table 1 is for a simple sanity check which agrees well.**

*Line 273.  I think the 28 % and 60% are the other way round.*

**Apologies. We will correct the text.**

*Lines 285 – 302.  I don't quite understand how this analysis is done.  Is it that you determine R for each week separately, and apply the R value for a given week.  Then calculate R for each 2 week period, and apply to all the measurements in that 2 week period.  Then repeat for longer and longer periods?  How is the uncertainty calculated?  From what is currently presented in the paper, I would interpret Figure 7 differently:*
*Figure 7 shows low uncertainty in $CO_2ff$ when a single week is used, and then a much larger uncertainty for all longer periods, with a gradual improvement in uncertainty (for the CO + $^{13}C$ method) as more weeks are averaged.*
*But when R is calculated for a single week, there are only one or two data points to constrain the R value, I think.  So I wonder if the low uncertainty calculated when only a single week is used is*

*artificial due to having little data to test against. The noise as R increases to 2, 3, 4, 5 weeks is most likely an artifact of having small datasets, and smoothing of the uncertainty as the temporal resolution of R increases makes sense.*

*Thus my reading Figure 7 is that determining R over a longer period is better than using short averaging periods, probably simply due to the small number of flask measurements available for short averaging periods. Further, I don't see any convincing evidence that R should change over periods of weeks or months, although R almost certainly has changed over years/decades as air quality controls have improved. Given the large uncertainties in CO₂ff derived from $^{14}C$ and the small number of flask measurements, variability in R over the short term is more likely explained by that uncertainty than by real variability in the R value.*

**The reviewer is correct about how we calculate the R, except that we use a moving window of 2N weeks for N weeks in x-axis. Uncertainty in $CO2_{ff}$ is calculated relative to $CO2_{ff}$ estimated from $\Delta^{14}CO_2$ using samples collected throughout a year. Therefore, the low uncertainty calculated for short averaging periods are not an artifact of having small datasets.**